# Anthropological perspectives on *Miyupimaatisiiun* and the integration of oral health in primary care in the Cree communities of Northern Quebec

**Richa Shrivastava**[1☯], **Roxane Campeau**[2☯], **Yves Couturier**[3‡], **Jill Torrie**[4‡], **Felix Girard**[1‡], **Bousquet Marie-Pierre**[5‡], **Elham Emami**[6☯]*

1 Faculty of Dentistry, Université de Montréal, Montreal, Quebec, Canada, 2 Faculty of Music, Université de Montréal, Montréal, Quebec, Canada, 3 School of Social Work, Université de Sherbrooke, Sherbrooke, Quebec, Canada, 4 Specialised Services Department, Cree Board of Health and Social Services of James Bay, Mistissini, Quebec, Canada, 5 Department of Anthropology, University of Montreal, Montreal, Quebec, Canada, 6 Faculty of Dentistry, McGill University, Montreal, Quebec, Canada

☯ These authors contributed equally to this work.
‡ These authors also contributed equally to this work.
* elham.emami@mcgill.ca

**Data Availability Statement:** All relevant data are within the manuscript.

## Abstract

The integration of primary oral health care has a pivotal role in improving oral health outcomes and providing accessible and affordable health care. This article contributes to the deep understanding of the cultural aspects of the integration of oral health into primary health care at an Indigenous health organization. Proceeding from a collaborative and interdisciplinary research project evaluating the integration of oral health care within primary care in Eeyou Istchee, this research is based on group discussions (6) and individual interviews (36) with 74 participants (care providers, administrators, and patients) held in four Eastern James Bay Cree communities. This study anthropologically explored participants' perceptions about primary health care conceptualizations, culturally based approaches, and experiences of oral care services at this organization using a "two-eyed seeing" Indigenous framework. The study identified three key factors related to the integration of primary oral health care: Cree perception of primary health and oral health care, cultural safety, and health provider–patient communication and the role of silence. Study findings reflected a dichotomy of perception of primary health care and the relevant units of care between the Cree structural and cultural perspective and the non-Cree professional perspective. The Cree people perceived "household" as a unit of care in comparison to non-Cree who viewed "health care services" as units of care. Our results also underline the role of cultural safety agents to address the needs for cultural competence and the role of silence as implicit cultural protocol. Our anthropological analysis illustrates the potential for increasing the level of appreciation for both users and workers in oral care in the future by ameliorating communication skills and intercultural knowledge.

**Funding:** This research has been funded by the Canadian Institutes of Health Research (Grant number: GI1-145123). http://webapps.cihr-irsc.gc.ca/decisions/p/project_details.html?appId=343763&lang=en.

**Competing interests:** The authors have declared that no competing interests exist.

## Introduction

Primary health care (PHC) has been reported as an effective strategy in reducing health inequalities as well as in meeting the needs of the population [1]. Within the concept of Primary health care (PHC), the Common Risk Factor Approach has been adopted by the World Health Organization as a health promotion and prevention strategy [2]. Updated with social determinants in 2012, the Common Risk Factor Approach emphasizes equitable and optimal health care that can be achieved by focusing on broader social and environmental factors including the cultural background of communities and organizations [2].

Anthropological approaches contribute to Primary health care (PHC) by emphasizing the dynamic efficiency of both cultural and structural aspects of a health care organization [3]. By contrast, a biomedical perspective would promote being in good health mainly to avoid or combat physical and mental diseases. The Cree concept of *miyupimaatisiiun* highlighted by the anthropologist Naomi Adelson (1992) refers to a broader perspective that could be translated as "being alive well" [4]. *Miyupimaatisiiun* considers not only physical health but also psychological, social, and political well-being and recognizes health within the context of land and identity. *Miyupimaatisiiun* embodies the Cree traditional ways of life as part of the Cree identity [4].

Based on this concept, the Cree Board of Health and Social Services of James Bay (CBHSSJB) organization decided to use the word *miyupimaatisiiun* [4] instead of "health" in communication with other bodies to introduce its programs and committees. This strategy first appeared in 1990 in an introductory guideline for non-Indigenous health professionals working among the Crees [5]. *Miyupimaatisiiun* is now entirely embedded at a structural level within the Cree Board of Health and Social Services of James Bay (CBHSSJB), including a strategic planning process entitled *IIyuu Ahtaawin Miyupimaatisiiun* Planning, and the name of the department under which local health services centres (Community *Miyupimaatisiiun* Centres) are operating [6, 7].

By endorsing the holistic concept of *miyupimaatisiiun*, the Crees of Eeyou Istchee have integrated a comprehensive and innovative approach towards implementation of the Common Risk Factor Approach in their PHC. In this context, the Cree Board of Health and Social Services of James Bay (CBHSSJB) has developed multiple programs such as Â Mashkûpimâtsît Awâsh. Â Mashkûpimâtsît Awâsh program delivers general services to families and young children (0–9 years) with the following goal: *To maximize the health and well-being of pregnant women, young children, and their families living in Eeyou Istchee. This goal has been achieved by providing a complete network of health, oral health, and social services. These services are grounded in Cree culture, values, and language, and rooted in family-friendly communities* [8].

The CBHSSJB develops its strategic regional plan as a planning tool [9]. The first strategic plan, for 2004–2014, was developed in 2004 with the aim of addressing the health and social services needs of the Eeyou Istchee population [10]. The subsequent second strategic regional plan for 2016–2021 guides continued development of health and social services in these communities with a particular focus on providing enhanced, culturally safe health and social services [9].

In establishing its PHC services the CBHSSJB has included the integration of oral health care [10–12]. This anthropological analysis study was conducted to have a deep understanding of the cultural aspects of the integration of oral health into PHC in this Indigenous health care organization.

## Materials and methods

### Setting and study design

The eastern James Bay region of northern Quebec in Canada comprises nine remote communities with a total population of nearly 18,000 Cree people of Eeyou Istchee [13]. The CBHSSJB

organization is responsible for the health and social services of these communities [10]. This anthropological analysis is nested into the comprehensive project entitled "Oral Health Integrated into Primary Care: Participatory Evaluation of Implementation and Performance in Quebec Cree Communities" [14]. This is a Canadian Institutes of Health Research funded participatory research project that involved an interdisciplinary team of researchers and professionals from Canada [14]. This comprehensive project commenced with a planning phase that included a planning meeting at one of the communities followed by a real-time video-conference workshop held at Montreal and Mistissini in 2016 involving an interdisciplinary team of Cree community members, administrators, and health and oral health care providers from Cree communities, public health representatives, as well as researchers [11, 12]. Ethics approval for this study was obtained from the Institutional Review Board of the Université de Montréal and McGill University and in addition permission was obtained from the Research Committee of the Cree Board of Health and Social Services of James Bay. Written informed and verbal consents were obtained from all the study participants. Our study adheres to the consolidated criteria for reporting qualitative research (COREQ) guideline for reporting qualitative studies [15].

## Data collection

A total of four different Cree communities out of the nine were purposefully selected for data collection. Maximum variation and snowball samplings were used to select key informants (health care providers, administrators and patients) for data collection. In-depth audio-recorded interviews and group discussions, on average 60 minutes long, were conducted in English or French by two research team members trained in qualitative methods. The data was collected in a quiet room at communities' local and regional hospitals. The interviews and group discussions were conducted using an interview grid with open-ended questions developed based on the Rainbow Model of Integrated Care [16, 17]. From the selected communities, a total of 74 participants including 46 health and oral health care service providers, 18 administrators, and 10 patients participated in 36 interviews and six focus group discussions. Among these participants, 43 were Crees and 31 were non-Crees [18].

## Data analysis

Verbatim transcripts of the interviews and focus group discussions were analyzed via a series of iterative readings. Within the major project's evaluation, this study was focused on the understanding of the cultural aspects of integration of oral health into PHC. We provide here an analysis of the verbatim transcripts using the Indigenous framework of "*Two-Eyed Seeing.*" This framework has been adopted as the analytic framework as it has been successfully applied to health anthropology research [19–23]. In this article, this framework helped in understanding the cultural aspect of integrated oral health in PHC at CHBSSJB. During this analysis, we focused on comprehending participants' definitions and perceptions of PHC and perception of culturally competent health services provision. We also compared findings across participants with special attention towards complementarities between Cree and non-Cree participants' perspectives. Data analysis was done manually as well as with the help of ATLAS.ti software (ATLAS-ti Scientific Software Development GmbH, Berlin, Germany). Two research trainees (RS, RC) independently conducted coding and thematic analyses and then discussed their analysis until consensus was reached. Subsequently, the analysis was revised by other members of the research team.

## Results

This study demonstrates three key factors associated with integration of primary oral health care at CBHSSJB: Cree perception of primary health and oral health care, cultural safety, and health provider–patient communication and the role of silence.

### Cree perception of primary health care and primary oral health care

Most of the Cree participants discerned PHC and primary oral health care as prevention, promotion, and interventions that are given in the communities via themselves as well as care providers in the health care organization. For instance, a Cree participant specifically mentioned that primary care was related to *taking care of yourself* (Participant 1, Interview). Also, the Cree participants working for the Â Mashkûpimâtsît Awâsh program specifically stated that PHC deals with providing health information and interventions starting before the baby's birth via involving pregnant women and then throughout the person's lifespan.

> *Primary care should start like from pregnant woman to the baby and up, . . . that is how I would look at it.* (Participant 2, Interview)

Similarly, non-Cree health professionals also emphasized being able to *take care of ourselves* as part of PHC. For example, a non-Cree health professional defined PHC as *to be able to fulfill his basic needs*. This participant identified PHC as the ability *to dress ourselves, to be ready for the next day, and to have good personal hygiene* as basic requirements (Participant 3, Interview).

Moreover, some participants, in a broader sense, interpreted PHC as holistic health care that include physical, emotional, mental, and spiritual components of health.

> *The primary care are the components necessary for an individual to be physically, emotionally, mentally, and spiritually fulfilled. So, shelter, food, love, peace, safety, some connection with nature, religious beliefs, and so on.* (Participant 4, focus group discussion)

> *Primary care is . . . the whole . . . . physical and spiritual thing, emotional . . . and . . . mental.* (Participant 5, Interview)

Concerning oral health, most participants considered dental health as an essential element of PHC and dental care providers as a major part of the PHC providers' team at CBHSSJB. Many participants (Cree and non-Cree) included everyone from the clinic to the community as PHC interveners who can contribute to fulfilling the aforementioned basic needs. These PHC interveners included doctors, nurses, dentists, dental hygienists, nutritionists, dental Cree support staff (secretary, dental assistant), ergotherapist, pharmacist, social workers, community health organizers, community health representative, home care workers, Cree Health Board managers, and so on. Nevertheless, most Cree participants considered parents as a prime PHC provider.

> *The parents play a role in primary oral health care because everything starts at home, it starts at home.* (Participant 5, Interview)

Along these lines, one Cree participant identified family members as PHC or primary oral health care providers since family members can *help you to love yourself* (Participant 6, focus group discussion). Moreover, Cree health professionals and para-professional participants characterized family members as indissociable and interdependent, with every member

responsible for caring for themselves and their siblings in the same ways. A Cree elderly participant also emphasized the role of parents and families while providing a Cree definition of primary care as follows: *Primary care is the parental responsibility, the word primary care, when you translate into Cree, it means, those who brought those people into this world are the primary caregivers, that's what it means, so parental responsibility is a key aspect to prevention and promotion for oral health* (Participant 7, focus group discussion). Likewise, another Cree participant described primary care as *Primary care is . . . just think of family, family are the first people that care for the others* (Participant 8, Interview).

Beyond the biomedical perspective, families are considered as essential as other health care providers by the participants. Furthermore, the majority of Cree participants pointed out that everything about PHC *starts from home* or *home is* central to PHC. However, most of the non-Cree participants referred to PHC as first-line care and emergency care.

## Cultural safety

Cultural safety is a principle, objective, and action plan adopted by the CBHSSJB. Its *Nishiyuu Miyupimaatisiiun* Department has the responsibility to ensure that Cree cultural realities and values are embedded in every service provided by the CBHSSJB, from strategic planning to the realization of programs [8].

> *The Cree health board is trying to integrate our Cree teaching, our Cree practices in our organization now. But it is coming slowly, [. . .] teachings, its almost . . . kind of coming back. I'll give you an example, we call it a Moss Bag (Aspahpisuwin). But it is where we put the baby inside, the newborn babies and you just wrap them up, and those are the kind of teachings that we are relearning. [. . .] So that's how I see culture right now being integrated.* (Participant 9, Interview)

A Cree participant referred to the trauma from the Residential Schools ("Canadian governments sponsored boarding schools to assimilate the Indigenous children into the Euro-Canadian culture" [24]) while discussing the integration of Cree traditions to oral health care.

> *It's hard for me to say really how we can do [to integrate better the Cree traditions with oral health care], like not knowing really the cultural. I have some aspects; I grew up in . . . the white society. I didn't grow up in my culture when I was young. I grew up in school. The school was my home, [. . .] I went to a residential school [. . .], then I moved to Chisasibi. [. . .] Then, after that, I moved to Gatineau, so I was not around my culture that much. [. . .] So it's hard for me to [. . .], I know some now as I grew up in the community. I know [. . .] from living with my parents a short while and during the summer. So . . . it's quite hard to [. . .] when you don't live in that culture. [. . .] It's a struggle for me . . . there's still a lot of things. After school . . . I started working. I worked for my dad, then I started with the health board about 30 years ago. [. . .] And from there, it's what always . . ., it's always works for me. [. . .] Well actually, I haven't learned the culture. I only learned once I came back from school, out of school and that was in the early 80s.* (Participant 10, Interview)

This participant was struggling to admit his/her lack of traditions regarding medicine and to recall the familial rupture he/she had to go through. The similar struggle among other Cree participants showed that the Residential School generation did not necessarily feel competent to determine exactly what is crucial to cultural competence beyond a general cultural sensitivity. The impacts of Residential Schools are undermining agency [25], harming the individual participant's autonomy.

One Cree grandparent discussed the traditional Cree ways of personal oral health care, such as use of charcoal, ashes, and black spruce tree gum. This grandparent pointed out the loss of these traditions from the time of the Residential Schooling system and later rapid socioeconomic development in these communities in the recent past. These issues resulted in rapid lifestyle and dietary habit changes leading to more health and oral health-related problems. Therefore, this participant raised concern about current highly prevalent oral health problems and associated oral health disparities in the Eeyou Istchee communities.

*I remember my grandparents and my parents, my aunties, my uncles, they had pure white teeth . . . And so I asked my elder what did you use? And they said, first of all, . . . we would all take that piece of charcoal. And then we would rub it on our teeth. And that took the tartar and everything and cleaned our teeth white . . . And there are times, the elders would take just a spoonful of the ashes, put it in the water and then to shrink it down. [. . . ] They also said . . . we were always told on a regular basis to go into black spruce tree, and . . . we would gather spruce gum all the time and constantly chew the spruce gum. [. . .] So then, black spruce water, pine water, and all the different trees were used for infection . . . they were very antiseptic, and . . . . anti-bacterial also. . . . So, we lost that part of our traditions. And it broke off at the time when we went to Residential School. So you end up with people that are proba-bly 50 years and younger that we're not influenced by the Residential School, and that's the population that went in the fast age of social and economic development for the Cree, and from there, our lifestyle and their diet changed very rapidly. [. . .] We knew nothing about chocolate . . . when we were children. [. . .] So those are the historical aspect of it. So now we have extreme access to junk food . . . walk in the corner store and all you see is junk food star-ing at you. It aberrates me, so the same thing in the communities, in the convenience stores, in the grocery stores. [. . .] I think it's a degenerative effect, and people do not understand that. Our oral health has deteriorated from one generation to the other. . . we have to deal with in relation to oral health.* (Participant 7, focus group discussion)

However, some Cree participants suggested that the situation is different for the youngest generation who experience traditional and modern health care as complementary forces instead of bringing back the traumatic rupture of the Residential Schools system. Some partici-pants consider this approach towards available possibilities for improvement. For instance, a Cree participant recounted how his/her child handles this deliberation:

*I'll give you an example of my daughter. She grew up in the community, but I had to send her out for school [in Montreal], so she grew up mostly knowing the non-natives' way of health. [. . .] If this didn't work, there's non-native medical ways of dealing with medical issues. She was able to look for other ways from the traditional side of the community. Any kind of medi-cation that she would use with her family, so she was able to adapt both sides. If she knew it didn't work, she would jump to the other side and look for another medicine, the traditional medicine approach.* (Participant 10, Interview)

Both Cree and non-Cree participants considered traditional treatments while considering cultural competence. They suggested that recognizing bush medicines would lead to a better relationship between the non-Cree medical or dental care providers and Cree patients, as it would give their patients a certain legitimacy.

*Well, I heard about some people . . . in the old days, . . . how they would treat their pain, and ya I think it's the time, we should consider getting more traditional medicine.* (Participant 11, Interview)

Furthermore, one participant summarized the struggles happening within households and communities towards better oral health. This participant positively believes in better health and oral health for the next generations over time.

*It's still complicated to live like our elders, they are really pushing hard that we keep our culture and keeping that culture also interferes with . . . what outside people are bringing here, so it's a lot to take in, it's a lot to take in, and it's not easy to, to be able to do all that right away. [. . .] I continue . . . we're on the right path. It's just time, time and you can't change someone in one night. You can't change the way people think in a week, you know. . . my children, my children's children will probably have much better dental hygiene than they had and what we had, sort of things . . . you know every generation is gonna be different, better.* (Participant 12, Interview)

## Cree provider–patient communication and role of silence

Cree participants pointed to workers' attitudes and professionalism, demanding to be treated the same way as "in the South," with a "more open-minded" and flexible attitude. Non-Cree participants confirmed this tendency by underlining that good communication can "open up" the patients and allows providers to connect with them, which in turn allows them to gradually become more active. A non-Cree health professional referred to communication as an investment that requires time and patience, that will lead to more proactivity from the patients, and that occurs gradually.

One significant result from our analysis is the identification of silence as a local implicit communication protocol that must be understood by outsiders. In this regard, a Cree participant explains in detail how communication is a struggle between Cree and non-Cree:

*You know . . . sometimes it's the communication that [. . .] plays a big role! [. . .] For example, when they ask questions or when they [Cree patients] respond, usually they don't respond by talking right away, it's almost like a complete silence first. And then they'll say something, a pause. . . . and sometimes it's the misinterpretation, maybe some non-native will say well, does he want to talk to me? But in reality, they're just kind of thinking before what they are saying . . . the pause.* (Participant 9, Interview)

## Discussion

PHC is defined as *the provision of integrated, accessible health care services by clinicians who are accountable for addressing a large majority of personal health care needs, developing a sustained partnership with patients, and practicing in the context of family and community* [26]. In Canada, the primary resources and agents in PHC are physicians and nurses [27]. From the dominant biomedical paradigm, PHC is aimed to deliver *whole-person care* for comprehensive health care needs throughout the life course. It is focused on the health care needs and preferences of individuals, families, as well as communities [28].

In this study, the anthropological analysis using a two-eyed seeing framework suggested a different understanding of units of care between Cree and non-Cree professionals. The unit of

care for Cree professionals was households, whereas that for non-Cree professionals was health care services. Our analysis also described the role of cultural safety agents in addressing the needs for cultural competence and the importance of silence during health provider–patient communications.

In regard to perception towards PHC, both Cree and non-Cree participants non-exclusively emphasized *taking care of ourselves*. This bridge between allochthonous and Indigenous perceptions about PHC signifies individual autonomy from a pragmatic perspective. However, this concept overlooks one of the essential Cree teachings inherited through Indigenous cultures, i.e., *taking care of ourselves* before being able to take care of others [25]. This Cree teaching is related to survival knowledge that relies on having belief in the animals' wisdom [25]. Animals are born with sufficient knowledge to assure their survival; they are able to take care of themselves, whereas humans have to learn it [29]. Learning from animals and other human beings is a lifelong process that one must do by her/himself.

Consequently, time and autonomy are highly significant in Cree culture. Moreover, all persons (animals are considered as non-human persons) have a specific autonomy [30] that corresponds to the way they interact with other beings reciprocally [31–35]. This complex meaning of individual autonomy allows us to look at particular dimensions of *taking care of ourselves*, depending on the person's social role. For instance, mothers' behavior has been identified as an active component of primary oral health care. The biomedical perspective might state that the mother's role as the primary oral health care provider is crucial since bacteria responsible for dental caries in infants are transmitted from mother to child [36]. Furthermore, the mother's role is aligned with the Â Mashkûpimâtsît Awâsh program's objectives because a pregnant woman will be considered autonomous by Cree people if she can take care of herself, thereby providing a healthy environment for the child in her womb. Another example is young adults' autonomy. Our results suggest that youth are autonomous because they are connecting with all available health spheres [37], and creating their internal experience-based model of health care. Thus, young people juggle biomedical aspects of health along with shared familial and social traditional knowledge [38].

Another dimension of *taking care of ourselves* that emerged from our results is the spiritual dimension of health. Spirituality encompasses the power to know, to love, and to will [39], which is progressively built into an individual during her/his life and within the family [30]. If spirituality is an essential component of PHC, then families are the spiritual support units. This suggests that family may be an irreducible foundation for conceiving the holding environment of PHC within Eeyou Istchee. However, we want to suggest that the household advantageously replaces family as an efficient way to refer to the people who are involved in a broader community-based definition of PHC. This terminology would assure that the concept of the family is not restricted to parent and children members, given that many participants point to propinquity to home and people who share the environment regularly as close family members. It emphasizes the Cree family-based social structure without imposing the nuclear family occidental model on it. It would also value the traditional conception of ecological collective homes described by Preston [40], enhancing the relational reliability that used to exist amongst people sharing a hunting territory. Nevertheless, the role of dentists, dental hygienists, dental assistants, and community health representatives in primary oral health care demonstrates the biomedical perspective of PHC, whereas many participants identified *households* as at least as important as hospitals in achieving effective primary oral health care.

Cree and non-Cree participants both want reliable services; however, Cree participants count on the household environment as a PHC resource. Borrowing from Hill [37], we can say that households are "open areas of decision-making" towards the best of what is offered to them. We need to clarify that we do not mean that there are individuals *and* households, but

individuals interacting within households. Here we are looking at collective autonomy that sustains individual autonomy [41]. From an anthropological ecological perspective, households as units of care belong to communities and society in general. Communities of the Cree Nation are not homogeneous: (1) there are coaster and inlander groups, which have their own particularities and specific hunting game, (2) there are northern and southern Cree Nations with northern and southern dialects and specific hunting practices, (3) within each communities, there are old and young people who are living very different realities, and finally, (4) there are the dynamic and cumulative characteristics of Cree society. As such, it would be misguided to consider in this analysis each individual as exhibiting either traditional or modern mainstream behavior [38]. Individuals inherit characteristics and teachings from their belonging people (Coasters/Inlanders, Northerners/Southerners, Old/Young generations). We must look at individuals as empowered by the accumulation of various traditional and biomedical knowledge acquired over time and through the institutions they build as a Nation. According to these considerations and our interpretations, decolonizing PHC necessitates recognizing the influence of households on individual health care behavior as eclectic and comprising advice about bush medicines and healing practices as well as relations to health care professionals and services based upon Western medicine. In this way, Cree and non-Cree participants involved in the primary oral health care can build from the best of their *Two Worlds*.

As the benefits of intercultural initiatives have been demonstrated in other cultural contexts [42], the development of cultural competence in Eeyou Istchee with a focus on providers' communication skills and opportunities to exchange knowledge bilaterally could improve the autonomy of patients and health professionals. Valuing Indigenous knowledge and local implicit protocols of communication may lead to an increase in trust and efficiency. Acknowledging respective strengths and adopting an "organizing concept of working across cultural collectives" such as the adoption of *miyupimaatisiiun* in the approach to health care services is a demonstration of socio-ecological resilience [43] that can contribute to the decolonization of health.

A better understanding of workers' and patients' narratives involves cultural competence [38], which has been proven as an effective component of PHC in specific populations, especially where there are issues related to ethnicity. In the Maori context, "the need for both the message and the messengers to belong to the same ethnic group has been emphasised" [44]. This element converges with the Truth and Reconciliation Commission of Canada, which underlines that hiring culturally aware workers and people who can speak Indigenous languages is part of the solution to increase the efficiency of health services in Indigenous communities [8]. The CBHSSJB is also working to develop cultural awareness training activities for workers. It has always given cultural leave time for Cree employees and has recently started to send non-Cree employees for workshops on the land in order to get hands-on experience and improve their cultural awareness [8]. It is well demonstrated in the literature that language barriers are related to accessibility of health care services [45] and while not specifically related to the Indigenous context, the importance of language also connects to political issues related to self-determination. It may be the co-presence of the accessibility issue and the self-determination objectives that characterize the situation in Eeyou Istchee. Language revitalization processes and valorization are indeed at the core of ongoing governance improvements that are happening within the Cree Nation.

The use of the *iiyiyuyimuwiin miyupimaatisiiun* links culturally and historically to the times when living on the land and relying on wildlife food was hard but worthwhile. Specifically concerning primary oral health care, elders recall that before widespread colonization, oral diseases were scarce. Few traditional medicines were known for tooth pain, and elders remember that people used to keep their teeth lifelong. Indeed, archeological evidence shows that oral

diseases were rare: less than 1% of the skulls of Arctic and North West Coast Indigenous populations found by archeologists have been reported with cavities [46]. Moreover, as per Mayhall, dental decay prevalence in the Canadian Arctic was originally very low and later increased gradually from the time of European contact and then rapidly after the intense contact with southern Canadian culture [47]. Oral care can consequently be seen as a burden from colonization, involving power issues. The integration of *miyupimaatisiiun* at the institutional level within the CBHSSJB and across the Cree communities begins to address these authority issues by affirming accountability within health care services. At the same time, if the primary oral health care providers advocate for changes in diet, stressing the consequences of the modern high-sugared diet, this recommendation poses inevitable challenges for Indigenous communities. Decolonizing diet is a difficult task since it involves modifying the daily life habits and avoiding the most affordable food (transformed and sugared) available in local markets. The diet's decolonization also requires that professionals such as nutritionists understand the value of bush food–*iiyimiichim*.

Since cultural competence focuses mostly on the education and the consciousness-raising of providers [48], improvement of their communication skills must be part of their training. Values are embedded in implicit protocols that regulate behaviors in a conventional way within every culture. Crees value keeping silence because silence gives the other the time to think about his/her reaction and about his/her input to the conversation or her/his reaction to an event, such as a diagnostic or a choice of treatment. Conversely, taking the time to pause before answering or reacting is seen as a mark of respect [49, 50]. Answering straightaway could also be interpreted as imposing one's view in an authoritarian manner [50]. Since coercion is culturally disapproved for the benefit of the development of observational skills and real-life experiences, silence is a demonstration of self-restraint and is thus valued. Moreover, verbal communication is relevant when there is an asymmetry of knowledge, and teachings must be given to others when they are ready to learn and when they are available to listen and be receptive. In primary oral health care services, this means that it should not be surprising to find that users of the services stay silent while seeking relief from pain caused by oral diseases.

Knowledge about health/illness exists in all old cultures and should be valued. Therefore, the thematic concepts derived and discussed in this study may not only have applicability to other similar Indigenous communities but also pertinent to the other countries. Our analysis was based on audio recordings and verbatim transcripts. Interviews and group discussions were not video recorded. However, video recording would have better supported our interpretation of the participants' involvement. Nevertheless, this anthropological research introduces a broader picture of the integration of oral health within PHC at the CBHSSJB and presents opportunities to pursue similar studies in other integrated health care settings.

## Conclusions

Walking on both shores of the biomedical and Indigenous ways of conceiving health, our results describe the variants of autonomy. First, PHC definitions revealed the strength of local structures and programs, the dynamic character of *iiyiyuyimuwiin* (Cree language) and self-governance institutions, highlighting Eeyou political and regional autonomy. Second, the PHC and primary oral health care perceptions enhance complementary views about the meaning of taking care of ourselves. Both lead to complementary versions of individual autonomy, from the individualistic perspective of non-Indigenous participants to the personal and community aspects implied within the discourse of Cree participants. Third, by revitalizing traditional knowledge and adapting it to actual structures and programs, the Cree Nation is improving collective autonomy. Lastly, our analysis points towards a local protocol that aims

for effective communication between users and providers in primary oral health care, which coincides with the linguistic and anthropological descriptions in the literature [49–51]. Our conclusions also converge with the strategic regional plans and the objectives of recent CBHSSJB annual reports [8–10, 52].

The CBHSSJB is continually working on *miyupimaatisiiun* and the integration of oral health in primary care in Eeyou Istchee. Nevertheless, it still needs time . . . time to develop valuable relationships between health professionals and the communities members; time to acknowledge the communication proficiency of the local protocols; and time to recognize that the patient will come to use the best of the Two Worlds in order to be alive well.

## Acknowledgments

Authors would like to express their sincere and profound gratitude to the Cree Board of Health and Social Services of James Bay, Cree communities, all the participants, and research team members for their constant help and support.

## Author Contributions

**Conceptualization:** Richa Shrivastava, Roxane Campeau, Elham Emami.

**Data curation:** Richa Shrivastava, Roxane Campeau, Elham Emami.

**Formal analysis:** Richa Shrivastava, Roxane Campeau, Elham Emami.

**Funding acquisition:** Elham Emami.

**Methodology:** Richa Shrivastava, Roxane Campeau, Elham Emami.

**Project administration:** Elham Emami.

**Resources:** Richa Shrivastava, Roxane Campeau.

**Supervision:** Elham Emami.

**Validation:** Richa Shrivastava, Roxane Campeau, Yves Couturier, Jill Torrie, Felix Girard, Bousquet Marie-Pierre, Elham Emami.

**Visualization:** Yves Couturier, Jill Torrie, Felix Girard, Bousquet Marie-Pierre, Elham Emami.

**Writing – original draft:** Richa Shrivastava, Roxane Campeau.

**Writing – review & editing:** Richa Shrivastava, Roxane Campeau, Yves Couturier, Jill Torrie, Felix Girard, Bousquet Marie-Pierre, Elham Emami.

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
