## [Decision Letter · Decision Letter 0]

24 Feb 2020

PONE-D-20-03395

‘It’s just time’: Anthropological perspectives on Miyupimaatisiiun and the integration of oral health in primary care in Eeyou Istchee

PLOS ONE

Dear Dr. Emami,

Thank you for submitting your manuscript to PLOS ONE. After careful consideration, we feel that it has merit but does not fully meet PLOS ONE’s publication criteria as it currently stands. Therefore, we invite you to submit a revised version of the manuscript that addresses the points raised during the review process.

We would appreciate receiving your revised manuscript by Apr 09 2020 11:59PM. To enhance the reproducibility of your results, we recommend that if applicable you deposit your laboratory protocols in protocols.io, where a protocol can be assigned its own identifier (DOI) such that it can be cited independently in the future. For instructions see: http://journals.plos.org/plosone/s/submission-guidelines#loc-laboratory-protocols

We look forward to receiving your revised manuscript.

Kind regards,

Lars-Peter Kamolz, M.D., Ph.D., M.Sc.

Academic Editor

PLOS ONE

Journal Requirements:

2. Please provide additional details regarding participant consent. In the ethics statement in the Methods and online submission information, please ensure that you have specified (1) whether consent was informed and (2) what type you obtained (for instance, written or verbal). If your study included minors, state whether you obtained consent from parents or guardians. If the need for consent was waived by the ethics committee, please include this information.

3. When reporting the results of qualitative research, we suggest consulting the COREQ guidelines: http://intqhc.oxfordjournals.org/content/19/6/349. In this case, please include more information on how participants were selected and recruited, and their characteristics.

4. We suggest you thoroughly copyedit your manuscript for language usage, spelling, and grammar. If you do not know anyone who can help you do this, you may wish to consider employing a professional scientific editing service.  

Reviewers' comments:

Reviewer's Responses to Questions

**Comments to the Author**

1. Is the manuscript technically sound, and do the data support the conclusions?

Reviewer #1: Yes

Reviewer #2: Yes

2. Has the statistical analysis been performed appropriately and rigorously? 

Reviewer #1: N/A

Reviewer #2: N/A

3. Have the authors made all data underlying the findings in their manuscript fully available?

Reviewer #1: Yes

Reviewer #2: Yes

4. Is the manuscript presented in an intelligible fashion and written in standard English?

Reviewer #1: Yes

Reviewer #2: Yes

5. Review Comments to the Author

Reviewer #1: Dear authors,

I got to review your manuscript „‚It‘s just time’: anthropological perspectives on Miyupimaatisiiun and the integration of oral health care in Eeyou Istchee“, in which you present your results of a qualitative investigative study about above mentioned subject.

First of all, I would like to congratulate you on this very well-written and interesting manuscript. In my opinion, it really ‚is about time‘, that ‚Western‘ health care practitioners open up for indigenous aspects and the other way round. Because of this aspect, I would like to suggest, to alter the title to „It‘s about time:...“ (if this correlates to what you want to say in the title).

Secondly, I would only recommend to present interviewees‘ demographics in the beginning of your results section table-wise. It would be interesting to know more detailed characteristics about age, origin, and so on to better interpret certain answers given.

Thank you.

Reviewer #2: Dear authors!

Thank you for the opportunity to review the manuscript, ‘It’s just time’: Anthropological perspectives on Miyupimaatisiiun and the integration of oral health in primary care in Eeyou Istchee”.

In principle, an absolutely interesting manuscript, and its publication is desirable. The basic statements could most probably apply to other contexts.

A few remarks:

The adaptation of the (full as well as short) title would be desirable. They should be worded more meaningfully.

Possibly use the abbreviations at least three times in the text without being abbreviated; that is, principally, only after that use abbreviations (e.g. line 63, 74, 113).

After very many – not necessarily customary – abbreviations are used, it is extremely useful to follow these recommendations and to, possibly, far beyond that continue to write out the used abbreviations (at various points in the manuscript).

Regarding of the importance of the subject, it would be helpful to adapt the keyword section.

The name of the authors should be cited according to the style guide.

Conclusion

… should contain an indication of the applicability of the thematic basic concepts to other countries.

Knowledge about health / illness exists in all (old) cultures and should be taken into account (e.g. concept of silence, Two-Eye seeing).

6. PLOS authors have the option to publish the peer review history of their article (what does this mean?). If published, this will include your full peer review and any attached files.

Reviewer #1: No

Reviewer #2: No

---

## [Author Response · Author response to Decision Letter 0]

5 Mar 2020

Response to Reviewers:

Reviewer #1: 

Comment 1. I got to review your manuscript „‚It‘s just time’: anthropological perspectives on Miyupimaatisiiun and the integration of oral health care in Eeyou Istchee“, in which you present your results of a qualitative investigative study about above mentioned subject. First of all, I would like to congratulate you on this very well-written and interesting manuscript.

Response: We would like to thank the reviewer for the time and constructive comments. 

Comment 2. In my opinion, it really ‚is about time‘, that ‚Western‘ health care practitioners open up for indigenous aspects and the other way round. Because of this aspect, I would like to suggest, to alter the title to „It‘s about time:...“ (if this correlates to what you want to say in the title).

Response: Thanks for the comment. We have removed the words ‘its just time’ from the title. We had taken this term from one of the participant’s quotations, where a participant is mentioning that time is needed to have good oral health status in the communities. Their health organization is working on this, but having good oral health can’t be achieved in a short period of time. 

We believe that putting ‘its just time’, there is a possibility of misinterpretation in the title. Hence considering reviews from both the reviewers about the title, we have changed the title now as: Anthropological perspectives on Miyupimaatisiiun and the integration of oral health in primary care in the Cree communities of Northern Quebec

Comment 3. Secondly, I would only recommend to present interviewees’ demographics in the beginning of your results section table-wise. It would be interesting to know more detailed characteristics about age, origin, and so on to better interpret certain answers given.

Response: We appreciate this comment. We have now added more information on participants’ characteristics. Please see page 6 (Line no. 122-125). As this study is a part is a bigger project, detailed participants’ characteristics have been presented in our previously published article under the same project. Now, we have included a reference for that article.

Reviewer #2: 

Comment 1. Thank you for the opportunity to review the manuscript, ‘It’s just time’: Anthropological perspectives on Miyupimaatisiiun and the integration of oral health in primary care in Eeyou Istchee”. In principle, an absolutely interesting manuscript, and its publication is desirable. The basic statements could most probably apply to other contexts. 

Response: We would like to thank the reviewer for the time and valuable comments.

Comment 2. The adaptation of the (full as well as short) title would be desirable. They should be worded more meaningfully.

Response: Thanks for the comment. We have changed the full title as: Anthropological perspectives on Miyupimaatisiiun and the integration of oral health in primary care in the Cree communities of Northern Quebec.

We adapted short title as: Anthropological perspectives on integrated primary oral health care in Cree communities. Please see page 1.

Comment 3. Possibly use the abbreviations at least three times in the text without being abbreviated; that is, principally, only after that use abbreviations (e.g. line 63, 74, 113). After very many – not necessarily customary – abbreviations are used, it is extremely useful to follow these recommendations and to, possibly, far beyond that continue to write out the used abbreviations (at various points in the manuscript).

Response: Thank you for the comment. We have now kept only two abbreviations, PHC (Primary health care) and CBHSSJB (Cree Board of Health and Social Services of James Bay). We have removed all other abbreviations from the manuscript. We have also kept the abbreviation in full for the first three times as advised.

Comment 4. Regarding of the importance of the subject, it would be helpful to adapt the keyword section.

Response: We have adapted the keyword section as follows: Indigenous health; Two-eyed seeing; Primary health care; Oral health; Anthropology; Cultural competence. Please see Page 2, Line no. 44-45

Comment 5. The name of the authors should be cited according to the style guide.

Response: Thanks for the comment. We have changed the citing authors in the text as per the style guide. Please see Page 18, Line no. 422.

Comment 6. Conclusion… should contain an indication of the applicability of the thematic basic concepts to other countries. Knowledge about health / illness exists in all (old) cultures and should be taken into account (e.g. concept of silence, Two-Eye seeing).

Response: We appreciate this comment. We have added this point as a concluding point in the discussion section. Please see. Page 19, Line no. 450-452.

---

## [Decision Letter · Decision Letter 1]

24 Mar 2020

Anthropological perspectives on Miyupimaatisiiun and the integration of oral health in primary care in the Cree communities of Northern Quebec

PONE-D-20-03395R1

Dear Dr. Emami,

We are pleased to inform you that your manuscript has been judged scientifically suitable for publication and will be formally accepted for publication once it complies with all outstanding technical requirements.

With kind regards,

Lars-Peter Kamolz, M.D., Ph.D., M.Sc.

Academic Editor

PLOS ONE

**Comments to the Author**

1. If the authors have adequately addressed your comments raised in a previous round of review and you feel that this manuscript is now acceptable for publication, you may indicate that here to bypass the “Comments to the Author” section, enter your conflict of interest statement in the “Confidential to Editor” section, and submit your "Accept" recommendation.

Reviewer #1: All comments have been addressed

Reviewer #2: All comments have been addressed

2. Is the manuscript technically sound, and do the data support the conclusions?

Reviewer #1: (No Response)

Reviewer #2: (No Response)

3. Has the statistical analysis been performed appropriately and rigorously? 

Reviewer #1: (No Response)

Reviewer #2: (No Response)

4. Have the authors made all data underlying the findings in their manuscript fully available?

Reviewer #1: (No Response)

Reviewer #2: (No Response)

5. Is the manuscript presented in an intelligible fashion and written in standard English?

Reviewer #1: (No Response)

Reviewer #2: (No Response)

6. Review Comments to the Author

Reviewer #1: (No Response)

Reviewer #2: (No Response)

7. PLOS authors have the option to publish the peer review history of their article (what does this mean?). If published, this will include your full peer review and any attached files.

Reviewer #1: No

Reviewer #2: No

---

## [Editor Report · Acceptance letter]

27 Mar 2020

PONE-D-20-03395R1 

Anthropological perspectives on Miyupimaatisiiun and the integration of oral health in primary care in the Cree communities of Northern Quebec 

Dear Dr. Emami:

I am pleased to inform you that your manuscript has been deemed suitable for publication in PLOS ONE. Congratulations! Your manuscript is now with our production department. 

With kind regards,

on behalf of

Dr. Lars-Peter Kamolz 

Academic Editor

PLOS ONE